# Focal Mechanism and Source Parameters Analysis of Mining-Induced Earthquakes Based on Relative Moment Tensor Inversion

**DOI:** 10.3390/ijerph19127352

**Published:** 2022-06-15

**Authors:** Anye Cao, Yaoqi Liu, Fan Chen, Qi Hao, Xu Yang, Changbin Wang, Xianxi Bai

**Affiliations:** 1School of Mines, China University of Mining & Technology, Xuzhou 221116, China; 4992@cumt.edu.cn (A.C.); haoqi@cumt.edu.cn (Q.H.); baixianxi@cumt.edu.cn (X.B.); 2Jiangsu Engineering Laboratory of Mine Earthquake Monitoring and Prevention, China University of Mining & Technology, Xuzhou 221116, China; 3State Key Laboratory of Coastal and Offshore Engineering, Dalian University of Technology, Dalian 116024, China; chenfansince95@163.com; 4School of Computer Science & Technology, China University of Mining & Technology, Xuzhou 221116, China; yang_xu@cumt.edu.cn; 5State Key Laboratory of Coal Resources and Safe Mining, China University of Mining and Technology, Xuzhou 221116, China; changbin.wang@cumt.edu.cn

**Keywords:** mining-induced earthquakes, relative moment tensor inversion, focal mechanism, source parameters, response laws

## Abstract

Mining-induced earthquakes (MIEs) in underground coal mines have been a common phenomenon that easily triggers rock bursts, but the mechanism is not understood clearly. This research investigates the laws of focal mechanism and source parameters based on focal mechanism and source parameters analysis of MIEs in three frequent rock burst areas. The relative moment tensor inversion (MTI) method was introduced, and the way to construct the inversion matrix was modified. The minimum ray and source number conditions were calculated, and an optimized identification criterion for source rupture type was proposed. Results show that the geological structure, stress environment, and source horizon influence the focal mechanism. The tensile type sources can distribute in the roof and coal seam, while the shear types are primarily located in the coal seam. In the typical fold structure area, the difference in source rupture strength and stress adjustment between tensile and shear types is negligible, while the disturbance scale of tensile types is distinct. The shear types have higher apparent volume and seismic moment in the deep buried fault area but lower source energy. The apparent stress of the tensile types is higher than that of the shear types, representing that the stress concentration still exists in the roof after the MIEs, but the stress near the faults could be effectively released. In the high-stress roadway pillar area, the primary fracture of the coal pillar easily produces a continuous shear rupture along the dominant stress direction under the extrusion of the roof and floor. The source parameters (except apparent stress) of shear types are higher than tensile types and have higher dynamic risk. The results contribute to expanding the understanding of rock burst mechanisms and guide MIEs’ prevention.

## 1. Introduction

Mining-induced earthquakes (MIEs) describe the physical–mechanical response of coal and rock mass to the normal consequence of operation in underground coal mines [1,2]. As the mining depth increases, geological engineering conditions have become more complicated [3]. Subsequently, the term rock burst triggered by MIEs has been revealed as a common dynamic disaster in underground mining, which usually leads to fatalities, equipment trouble, and roadway damage with the release of lots of elastic energy [4,5,6,7,8]. The mechanism of rock burst is still one of the least understood and challenging tasks of underground operation in many countries, including the USA, Australia, China, Poland, Russia, South Africa, Germany, and Canada [9,10].

Recently, rock burst triggered by MIEs has been reported in many kinds of geological conditions, including fold structure, fault area, pillar area, etc. For example, the “10.20” rock burst occurred in the Longyun Coal Mine (21 people killed and 4 people injured) triggered by two combined faults [11]. In addition, 9 June 2019 witnessed 9 people being killed and 12 being injured underground in the Longjiapu Coal Mine [12]. Six rock bursts continually occurred during the retreat of the LW401102, a typical burst-prone longwall face in the fold structure area [13]. Similarly, a severe rock burst was reported in the isolated coal pillar area [14]. Numerous methods and efforts have been made to acquire a comprehensive understanding of the rock burst mechanism, including physical and similar simulation experiments [15,16], numerical modeling [17,18], laboratory tests [19,20], and field investigation [21,22], etc. Therein, numerical modeling also sheds light on the application of assessing rock burst risk [8,17,23,24]. For example, the stress, displacement, and energy evolution characteristics along the rock burst process can be investigated in detail. However, the rupture mechanism of MIEs and the relationship between source parameters and rock burst has not been systematically investigated due to unreliable methods.

Since MIEs are closely related to the stress and strain of rock mass, routine seismic monitoring has been incorporated into rock mechanics and is one of the primary means to investigate and quantify the mechanical response to underground operations [25]. Currently, several seismic-based methods have been developed to understand rock response, including seismic energy analysis [26,27,28], source parameters [29,30], focal mechanism [31,32,33], ground motion [34], etc. Especially for focal mechanism, moment tensor inversion (MTI) has been proved a powerful tool to understand the source mechanism, which can characterize the rock’s motion state and rupture mode [35]. Recently, several MTI methods have been developed through fitting different kinds of observations and source models, including seismic waveforms [36], first motion polarities [37], P- and S-wave amplitude ratios [38], the combination of spectral amplitudes or amplitude ratios with P-wave polarities [39] and full-waveform method [40], etc. Therein, the accurate construction of Green’s functions is necessary. However, as the MTI of these methods is influenced by station layouts, focal mechanism, medium properties, etc., when performing on MIEs in coal mines [41], the accurate construction of Green’s functions is complicated. Therefore, a modified MTI method applying to MIEs in underground coal mines should be explored to understand the source mechanism of MIEs.

This paper conducted a focal mechanism and source parameters analysis of MIEs based on relative moment tensor inversion to systematically understand the rock burst characteristics in three frequent rock burst areas with different geological features. After comparing the inversion principle of the absolute and relative MTI method, an inversion method of relative moment tensors to optimize the inversion matrix construction method was proposed to obtain the focal mechanism solution. The inversion matrix was constructed based on the source group, and the source rupture type identification method was optimized. Subsequently, the response laws of focal mechanism and source parameters of MIEs in three frequent rock burst areas were analyzed.

## 2. Inversion Method of the Relative Moment Tensor for the Focal Mechanism of Mining-Induced Earthquakes

Based on the arrangement of seismic monitoring systems in coal mines, the focal mechanism and source parameters of MIEs can be calculated to evaluate the dynamic hazard [5]. Along with the elaboration of the source theory, people have realized that the source mechanical state at the rupture time is more complex than simply measuring it by a single force or a pair of staggered forces [42]. Therefore, the MTI is introduced to explain the rupture behavior of sources. For example, the response characteristics and rupture types of key strata during periodic tests in a Chinese coal mine were identified via the MTI [35].

### 2.1. Purpose of Focal Mechanism Inversion via MTI

The purpose of focal mechanism inversion is that people hope that the stress state at the time of source rupture can be grasped to analyze the mechanism of source rupture better. When the earthquake wavelength is much smaller than the distance between the source and the station in space and the period of earthquake wave is much longer than the rise time of the earthquake wave in temporal, the earthquake sources can be regarded as point sources [41]. For a point source, in the Cartesian coordinate system, the moment tensor of sources ***M*** can be expressed as:(1)$$M=\left[\begin{array}{ccc} M_{11} & M_{12} & M_{13}\\ M_{21} & M_{22} & M_{23}\\ M_{31} & M_{32} & M_{33} \end{array}\right]$$
where $M_{ij}\left(i,j=1,2,3\right)$ represents the value of the moment tensor and the first and second subscript represents the axial direction of the force couples and the direction in which it points. The nine different force couples of the moment tensor are depicted in Figure 1. Due to the moment tensor meeting the condition that angular momentum is conserved (e.g., $M_{ij}=M_{ji}$), the ***M*** has only six independent elements [43].

The relationship between the monitored displacement $u_{ij,k}^{n}(x,t)$, Green’s function $G_{ij,k}^{n}(x,x_{0},t)$, and the moment tensor $M_{jk}$ can be expressed as Equation (2), which will be used to invert the moment tensor of the point source.
(2)$$u_{ij,k}^{n}(x,t)=G_{ij,k}^{n}(x,x_{0},t)M_{jk}*S(t)\begin{array}{cc} & i,j,k,n=1,2,3 \end{array}$$
where the $u_{ij,k}^{n}(x,t)$ means the displacement of the *i*-th source caused by the *k*-th seismic phase in the *n*-th direction of the *j*-th station, here *k* = 1, 2, or 3 indicates the waves excited from the *P*, *SH*, and *SV* seismic phase, respectively, and *n* = 1, 2, or 3 represents three orthogonal directions; $G_{ij,k}^{n}(x,x_{0},t)$ represents the relationship between the unit force at time *t* and point *x_0_* and the displacement produced at point *x*; the symbol “∗” is the convolution operation, and the S(t) represents the time function of source. It can be noticed that S(t) is replaced as an impulse function δ(t) due to the synchrosource hypothesis (i.e., the period of earthquake wave is much longer than the rise time of the earthquake wave in temporal.). Especially for MIEs (i.e., the medium and small earthquakes), the earthquake wave propagation time is approximately zero, so the impulse function δ(t) is nearly equal to 1 [44]. Therefore, Equation (2) for MIEs can be simplified as:(3)$$u_{ij,k}^{n}(x,t)=G_{ij,k}^{n}(x,x_{0},t)M_{jk}\begin{array}{cc} & i,j,k,n=1,2,3 \end{array}.$$

Furthermore, Equation (3) can be expressed in matrix form as:(4)u=GM
where ***u*** is the far-field displacement sequence matrix obtained from different stations, which means that only the displacement sequence obtained from stations far enough from the source Equation (4) can be constructed, and the minimum displacement condition is proposed as more than 500 m in underground mining [45]; ***G*** is the theoretical Green’s function related to the velocity model of formation structure.

It is widely accepted that the low-frequency spectral level $\Omega_{0}$ can be regarded as the displacement $u_{i}$, which can be calculated as [46]:(5)$$\left\{\begin{array}{c} S_{D2}=2\int_{0}^{\infty }D^{2}(t)dt\\ S_{V2}=2\int_{0}^{\infty }V^{2}(t)dt\\ \Omega_{0}=2\sqrt{4S_{{}_{D2}}^{3/2}S_{{}_{V2}}^{-1/2}} \end{array}\right.$$
where $S_{D2}$ and $S_{V2}$are the squared spectral displacement and velocity integrals for both P- and S-waves. In addition, the direction of $\Omega_{0}$ is judged according to the relative location between MIEs and stations, which can be expressed as [47]:(6)$$\left\{\begin{array}{l} \mathrm{Positive}\text{,}\mathrm{The}\;\mathrm{sensor}\;\mathrm{located}\;\mathrm{above}\;\mathrm{the}\;\mathrm{source},\;\mathrm{and}\;\mathrm{the}\;\mathrm{waveform}\;\mathrm{initially}\;\mathrm{moved}\;\mathrm{up}\\ \mathrm{Negative}\text{,}\mathrm{The}\;\mathrm{sensor}\;\mathrm{located}\;\mathrm{above}\;\mathrm{the}\;\mathrm{source},\;\mathrm{and}\;\mathrm{the}\;\mathrm{waveform}\;\mathrm{initially}\;\mathrm{moved}\;\mathrm{down}\\ \mathrm{Positive}\text{,}\mathrm{The}\;\mathrm{sensor}\;\mathrm{located}\;\mathrm{below}\;\mathrm{the}\;\mathrm{source},\;\mathrm{and}\;\mathrm{the}\;\mathrm{waveform}\;\mathrm{initially}\;\mathrm{moved}\;\mathrm{down}\\ \mathrm{Negative}\text{,}\mathrm{The}\;\mathrm{sensor}\;\mathrm{located}\;\mathrm{below}\;\mathrm{the}\;\mathrm{source},\;\mathrm{and}\;\mathrm{the}\;\mathrm{waveform}\;\mathrm{initially}\;\mathrm{moved}\;\mathrm{up} \end{array}\right.$$

Therefore, the goal of MTI is acquiring the ***M*** according to Equation (4) and decomposing it to further explain the source rupture mechanism. The MTI can be divided into absolute, relative, and mixed inversion methods according to the treatment of Green’s function [48].

Figure 2 describes the schematic diagram of the absolute MTI method. In this method, Green’s function for each event is calculated based on the relationship between source location, station layout, and attenuation model of medium [41]. Hence, the inversion accuracy of the absolute MTI method mainly depends on Green’s function. Although the hybrid moment tensor method reduces the influence of noise through repeated iteration, the inversion theory is the same as that of the absolute method. However, due to the complex geological environments and possible lateral inhomogeneities, it is always difficult to calculate Green’s function accurately [48]. The relative MTI method was adopted to solve this problem since Green’s function for each event was not calculated in the process.

### 2.2. Application and Optimization of Relative Moment Tensor Inversion Method in Underground Coal Mines

Figure 3 describes the schematic diagram of the relative MTI method. In this method, they assume that the sources of the cluster experience a similar ray-path or propagation process, and the Green’s function of the sources from the same source region is common. Generally, a reference event whose radiation pattern is known is used to estimate Green’s function [49,50,51]. However, it is challenging to find a perfectly proper reference. Luckily, the relative MTI method proposed by Dahm [52] can overcome this shortcoming, i.e., the reference source is not needed anymore. Green’s functions are simplified in his method as linear wave propagation terms and radiation patterns based on the ray-theoretical limit. The linear wave propagation terms are eliminated when radiation patterns are known—thereby avoiding the explicit calculation of Green’s function.

It is emphasized that the relative MTI method has two significant advantages over the absolute one: (1) it avoids solving complex Green’s function and greatly improves the inversion accuracy; (2) the inversion efficiency is greatly improved by using source cluster compared with single-source inversion in absolute MTI. However, when introducing the relative MTI in the source mechanism of MIEs, several difficulties must be solved: (1) how to select the appropriate source cluster is necessary since MIEs are randomly distributed in temporal and space, i.e., the minimum ray and source number conditions should be calculated; (2) the construction of the inversion matrix needs to be optimized under the background of the stations in coal mine move along with the excavation in contrast to Dahm’s; (3) the constructed of the inversion matrix, which may be a large sparse matrix, needs an effective solution method. The detailed description of the relative method inversion matrix construction and solution method can be seen in Appendix A and Appendix B, respectively.

## 3. Focal Mechanism Solution of MIEs

The focal mechanism solution is a prerequisite for analyzing the seismic source rupture mechanism by moment tensor theory. In detail, the rupture type and the occurrence of the rupture surface (i.e., azimuth, dip and slip angle) can be determined by decomposing the moment tensor. In addition, many applications of moment tensor theory are derived from these parameters. For instance, Šílený and Milev [53] distinguished roof caving, roof rupture, coal pillar instability, and low subduction reverse fault type fracture modes by inverting the focal mechanism of five mining-induced earthquakes in the Driefontein gold mine in South Africa. Moreover, several typical applications of MTI of mining-induced earthquakes have been developed, including the inversion of the rupture propagation process in hydraulic fracturing [53,54], the guidance of hydraulic fracturing parameter design [55], mining-induced stress field inversion, [56,57] etc. In this process, the effectiveness of inversion analysis depends on the suitability of the focal mechanism solution method.

### 3.1. Solution of Source Rupture Occurrence

According to the source rupture theory, a general dislocation model (seen in Figure 4) can describe the source rupture occurrence. The relationship between moment tensor component and rupture surface occurrence is as follows:(7)$$\begin{array}{ll} {\bar{M}}_{11}=-\kappa \cos\alpha (\sin2\phi \sin\theta \cos\gamma +\sin^{2}\phi \sin2\theta \sin\gamma ) & +\sin\alpha (1+2\kappa \sin^{2}\phi \sin^{2}\theta )\\ {\bar{M}}_{12}=+\kappa \cos\alpha (\cos2\phi \sin\theta \cos\gamma +0.5\sin2\phi \sin2\theta \sin\gamma ) & -\kappa \sin\alpha \sin2\phi \sin^{2}\theta \\ {\bar{M}}_{13}=-\kappa \cos\alpha (\cos\phi \cos\theta \cos\gamma +\sin\phi \cos2\theta \sin\gamma ) & +\kappa \sin\alpha \sin\phi \sin2\theta \\ {\bar{M}}_{22}=+\kappa \cos\alpha (\sin2\phi \sin\theta \cos\gamma -\cos^{2}\phi \sin2\theta \sin\gamma ) & +\sin\alpha (1+2\kappa \cos^{2}\phi \sin^{2}\theta )\\ {\bar{M}}_{23}=-\kappa \cos\alpha (\sin\phi \cos\theta \cos\gamma -\cos\phi \cos2\theta \sin\gamma ) & -\kappa \sin\alpha \cos\phi \sin2\theta \\ {\bar{M}}_{33}=+\kappa \cos\alpha \sin2\theta \sin\gamma & +\sin\alpha (1+2\kappa \cos^{2}\theta ) \end{array}$$
where ${\bar{M}}_{ij}=M_{ij}/M_{0}$ (*i*, *j* = 1 to 3); $M_{ij}$ represents the moment tensor without the isotropic components; $M_{0}$ represents the scalar seismic moment (formula for the calculation is shown in Appendix A); *κ* represents the ratio of Lame constant at the rupture surface (i.e., κ=λ/μ); λ and μ represent the first and second Lame constant, respectively; ϕ, θ, and γ represent the strike, dip, and slip angle, respectively; α represents the dislocation angle.

Equation (7) contains six nonlinear equations with five unknowns. Therefore, the problem of solving these equations can be transformed into searching for the best position in the space to make it closest to the target in the five-dimensional space. Meanwhile, the five unknowns have a clear range, which makes the range of five-dimensional space limited, and reduces the difficulty of solving the problem and the possibility of multiple solutions to a certain extent. Therefore, Particle swarm optimization (PSO) [58] was adopted here to solve this question.

### 3.2. Identification of Source Rupture Type

For identification of the rupture type, there are two widely accepted approaches: proportion of moment tensor components- and dislocation angle of rupture surface (seen in Figure 4)-based. For the former, the moment tensor component could be divided into a double-couple ($M_{\mathrm{DC}}$), isotropic component ($M_{\mathrm{ISO}}$), and a compensated linear vector dipole component ($M_{\mathrm{CLVD}}$), whose proportion adds up to 100% (i.e., $M_{\mathrm{DC}}\%+M_{\mathrm{ISO}}\%+M_{\mathrm{CLVD}}\%=100\%$) [43]. Consequently, [60] proposed to quantify the rupture type by the proportion of shear components ($M_{\mathrm{DC}}\%$), in which $M_{\mathrm{DC}}\%\geq 60\%$ is defined as shear type rupture, $M_{\mathrm{DC}}\%\leq 40\%$ is defined as tensile type rupture, and $40\%<M_{\mathrm{DC}}\%\text{<}60\%$ is a mixed-type rupture. For the latter, the dislocation angle of the rupture surface was proposed by Vavrycuk [61] as the basis for judging the rupture type. However, Ohtsu and Vavrycuk proposed the corresponding discriminant criteria for acoustic emission events and natural earthquakes, respectively. Due to the complexity and diversity of MIEs, if these criteria are applied to the classification of rupture type of MIEs, there may be a bias in the accurate identification of rupture type. Therefore, which parameters and ranges apply to the identification of MIEs’ rupture types should be further investigated.

In this section, to obtain the appropriate criterion of source rupture type for MIEs, a parametric discussion about the proportion of shear ratio in moment tensor components (**M**_DC_ %) and dislocation angle of rupture surface was conducted.

In source rupture theory, both the moment tensor components and occurrence of the rupture surface are related to the Lame constant of the medium, where the Lame constant is the material parameter characterizing the deformation of the medium in the strain–stress relationship (the detailed information is shown in Equation (7)). Therefore, it is indispensable to discuss the relationship between the **M**_DC_ % and dislocation angle (α) under the different proportions of the Lame constant (i.e., different μ/λ).

Figure 5 depicts the relationship between M_DC_ % and cosα for different μ/λ. The following results can be obtained:
There is a positive correlation between the **M**_DC_ % and cosα, which means that the two indexes are consistent in identifying the source rupture type, but both need a reasonable range of identification.There is a nonlinear correlation between them, indicating that the shear displacement is still dominant when the **M**_DC_ % is relatively low (**M**_DC_ % ∈ [20,40]). It reminds us that there may be a bias on those sources characterizing low **M**_DC_ % but high shear displacement.Due to the nonlinear relationship between **M**_DC_ % and cosα the specific interval of both **M**_DC_ % and cosα should change with μ/λ.


**Figure 5 ijerph-19-07352-f005:**
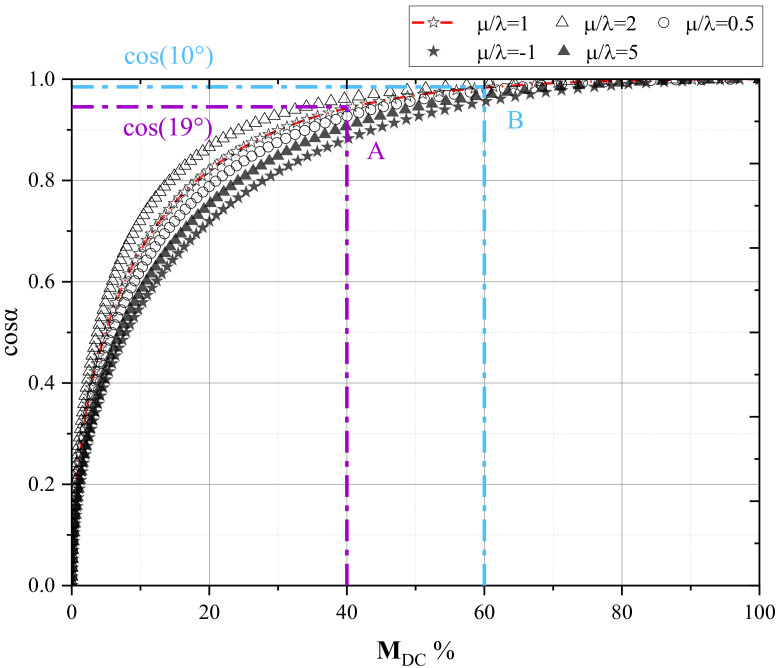
The relationship between **M**_DC_ % and cosα for different μ/λ.

It follows from the above statement that when the research object is a natural tectonic earthquake or rock specimen with pre-existed cracks, it is relatively efficient to identify the rupture type by **M**_DC_ %. However, MIEs may have more complex rupture mechanisms, including shear, tension, compression, implosion, and other fractures [62]. It can be noticed that the Lame constants must be taken into account when identifying source rupture types based on not only **M**_DC_ % but dislocation angle. The Lame constants are too abstract to apply to the rock mechanism. Therefore, the uniaxial compressive, tensile, and shear strength widely used in rock mechanics are used to replace Lame constants [45].

For extended-source events (dislocation angle larger than zero), the dislocation forms are shear dislocation and tensile dislocation, and the corresponding detection criteria for dislocation angle are as follows:(8)$$\tan\alpha >\frac{F_{t}}{F_{s}}.$$

For the shrinking-source event (the dislocation angle is less than zero), the dislocation form is shear dislocation and compression dislocation, and the corresponding detection standard of the dislocation angle is as follows:(9)$$\tan\alpha <\frac{F_{n}}{F_{s}}$$
where $F_{n}$, $F_{s}$, and $F_{t}$ are the sources’ uniaxial compressive, tensile, and shear strength. Suppose the mechanical properties of sources are not measured. In that case, the hypothetical relationship among them (i.e., the compressive strength of coal and rock mass is three times the shear strength and ten times the tensile strength) is proposed [45]. Consequently, the source rupture criterion combined with dislocation angle and Lame constants can be expressed as:(10)$$\left\{\begin{array}{l} \alpha \in (14{}^{\circ},90{}^{\circ}]\\ \alpha \in (-72{}^{\circ},14{}^{\circ})\\ \alpha \in [-90{}^{\circ},-72{}^{\circ}) \end{array}\right.\begin{array}{c} \mathrm{Tensile}\\ \mathrm{Shear}\\ \mathrm{Compress} \end{array}.$$

It is emphasized that if the mechanical properties of rock near the source have been tested, the criteria for identification of rupture types for MIEs should be constructed according to Equations (8) and (9).

### 3.3. Guidelines on the Implementation

Figure 6 shows the flowchart for implementing the method to apply. First, we should confirm the sources to be retrieved and the corresponding station information, including the three-dimensional coordinates of the sources, the station coordinates, and the wave velocity near the stations. Second, we should judge whether the sources satisfy the far-field condition. If meeting the far-field condition, the corresponding ray between source and station is valid; if not, those rays should be eliminated. Third, we should judge whether the screened conditions of source and ray numbers (seen in Equation (27)) are satisfied. Provided the two conditions are met, the inversion matrix can be constructed; if not, we should increase the number of sources involved in the inversion return to the first step.

After constructing the inversion matrix, we can solve it by Equations (28)–(31) and further decompose the moment tensor. Furthermore, the rupture occurrence (using Equation (7)), source rupture types (using Equation (10)), and beach balls can be obtained.

## 4. Response Laws of Focal Mechanism and Source Parameters (RLFMSPs) in Frequent Rock Burst Area

The focal mechanism can reveal the type and source of rupture occurrence. However, in the process of monitoring hazards induced by MIEs, the intensity information of source rupture, the scale of disturbance, and the level of stress change should be further understood to conduct targeted prevention. In addition, to further reveal the mechanism of MIEs and improve the ability to identify catastrophic events, the relationship between focal mechanism and source parameters should be analyzed in detail. Therefore, in this section, we compare and analyze the response laws of focal mechanism and source parameters (abbreviated as ‘RLFMSPs’ in this paper) in three different frequent rock burst mining areas in China. Therein, the Brune model [63] was used to calculate the source parameters, including scalar seismic moment, source radius, apparent volume, stress drop, and apparent stress. The physical meanings and formula for the calculation are listed in Appendix C.

### 4.1. A Case of RLFMSPs in Fold Structure Area

Huating Coal Mine, a typical burst-prone coal mine, is located in the southeast of Gansu Province, China. The first case study site, longwall (LW) 250105−1, is the upper stratified panel of No.5 coal seam with a length of 2000 m and a width of 200 m. There are 6 m rib block coal pillars between longwall panels to isolate the goaf. The mining depth of No.5 coal seam in LW250105−1 is 500~700 m, and the dip angle is between 1~5°. Figure 7a shows that the heterogeneous fold structure leads to a complex stress environment and higher local horizontal stress. The west of LW250105−1 is the goaf of LW250103−1 (seen in Figure 7c), and the east is solid coal mass. There are 6 m rib block coal pillars between longwall panels to isolate the goaf, and the stratigraphic column of LW250105−1 is shown in Figure 7b.

Since 2008, The Huating Coal Mine has been equipped with a seismic monitoring system “SOS” developed by the Central Institute of Mining in Poland to monitor seismicity related to mining activities continuously. As shown in Figure 7c, aiming to monitor the seismicity of LW250105−1, two types of stations are installed underground, i.e., entry stations and remote stations. The entry stations 1#, 2#, 7#, 13#, and 16# were installed in the longwall entries and moved regularly with the longwall retreat, and the other remote stations were placed in the roadway at different heights 2~5 km from the floor to cover the monitoring area. It should be emphasized that as a similar seismic system is used to monitor MIEs in the mining area in the later section, more details about the system and layouts are not mentioned.

There were 34 strong MIEs that occurred in LW250105−1 during the retreat from March 2014 to May 2015. The typical seismic waves of strong tensile type MIEs in LW250105−1 are depicted in Figure 8. We can see that the strong MIE caused several channels (such as channels 1#, 2#, 13#, and 16#) to exceed the limit, and the waveform detected by other stations far from the source lasted longer and attenuated slowly, which indicates that the strong MIE is characterized by strong high disturbance. In addition, the signal has poor P, SH, and SV amplitudes in some panels, e.g., panels 1#, 13#, and 16# in Figure 7. However, in the moment tensor inversion, these panels are not used, and only panels 2#, 3#, 4#, 5#, 6#, and 8# were used.

Figure 9a shows the source location and focal mechanism solution of 34 strong MIEs during the LW250105−1 retreat. There were 20 tensile type ruptures and 14 shear-type ruptures, indicating that the tensile-type ruptures were dominant among the 34 MIEs investigated. Moreover, the tensile rupture is mainly located in the roof and coal seam, while the shear rupture is primarily located near the coal seam. It can be seen that the separation and tensile rupture of the high and low roofs are the main reasons for the strong tensile type MIEs, and the buckling and shear failure of the floor under high-level tectonic stress is the main factor of the strong shear-type MIEs triggered by complex fold structure and mining disturbance.

Figure 9b depicts the source rupture occurrence of 34 strong MIEs. The dip angle of most sources with tensile type rupture is less than 30°, while the dip angle of those with shear type rupture is more than 30°, which indicates that strong MIEs are closely related to the rupture type. In addition, the dip angle of the rupture plane located on the floor is more than 60 degrees, and the tensile type sources on the roof have more rupture dip angles than those located on the floor, which indicates that the occurrence of the rupture plane of the strong MIEs is closely related to the source horizon.

Figure 9c displays the focal mechanism and response characteristics of the focal parameters of 34 strong MIEs. There are a few differences in source energy between the tensile-type and shear-type ruptures, which indicates almost no difference between the two-type ruptures in source intensity. Although both the tensile-type and shear-type sources have relatively close source radiuses, the apparent volume of the tensile types is 1.28 times that of the shear types. It showed us that the tensile types in the fold structure area have a larger disturbance scale. The apparent stress difference between tensile-type and shear-type ruptures is negligible, while the stress drop of shear types is about 10% higher than that of tensile types. It can be concluded that the rupture type of strong MIEs has little influence on the stress adjustment at the source.

### 4.2. A Case of RLFMSPs in Deep Buried Fault Area

The second case study site, LW3302, is located in the middle of Shandong Province, China. The LW3302 is fairly deep at about 1200 m underground with high in situ stress. In addition, as shown in Figure 10, a fault with a drop ranging from 1.4 m to 3.0 m and a dip of 60 degrees extended across the middle of the LW3302. During the LW3302 retreat between June and July 2015, 35 strong MIEs were detected near the LW3302, and seven rock bursts were induced in this process. The rock burst distribution is shown in Figure 10. It can be inferred that the existence of the fault not only led to the frequent MIEs and induced higher dynamic disaster but only high in situ stress.

Figure 11a shows the source location and focal mechanism solution of 35 strong MIEs during the LW3302 retreat between June and July 2014. The sources are mainly distributed in coal seams or low-level roofs on elevation, and parts are located on the high-level roof. Among them, the sources located in the roof are judged as tensile-type ruptures, while the sources that occurred in coal are mainly shear-type ruptures. It shows that when mining near the deep buried fault area, the strong mining disturbance of the longwall is easy to induce fault activation, and the coal and rock mass near the fault is easier to produce slip rupture, and the shear component is dominant in the moment tensor. Meanwhile, fault activation can easily lead to roof breaking and instability characterized by tensile type rupture.

Figure 11b depicts the source rupture occurrence of 35 strong MIEs in LW3302. It can be seen that the rupture azimuth of the sources is about 150° or 300° and approximately parallel to the mining direction of the LW3302. In addition, the dip angle of most sources is less than 40°, and only a few source dip angles are larger than 60°, which indicates that the occurrence of MIEs is closely related to the mining direction of longwall face and fault orientation during mining in deep fault areas.

Figure 11c shows the focal mechanism and response characteristics of focal parameters of 35 strong MIEs. The source energy of tensile-type sources is significantly higher than that of shear-type sources, while the seismic moment of shear-type sources is higher than that of tensile-type sources. Compared with the shear-type sources, the source energy radiating from sources in the roof will experience less attenuation and lead to higher dynamic risk. Furthermore, as the coal and rock mass in the fault area is relatively fractured, the fault will produce larger shear dislocation, which results in the seismic moment of shear types being higher than that of tensile types.

### 4.3. A Case of RLFMSPs in High-Stress Roadway Pillar Area

The third case study site, the roadway pillar area, is located in the south–middle of Shanxi Province, China. The operation target is the No.4 coal seam with an average thickness of 9.43 m and a buried depth of 800~1000 m. Due to the large buried depth, fold structure, and large goaf area around the roadways, special static stress is applied on the roadway pillars and leads to obvious stress concentration. Fifty strong MIEs were monitored from August to October 2018, and there were five rock bursts being induced (seen in Figure 12).

As seen in Figure 12 and Figure 13a, the coal pillars of the roadway bear a high abutment pressure due to the goaf area around the roadway. In addition, the repeated disturbance of the longwall mining leads to the development of primary fractures in the coal pillars, leading to a decrease in the stability of the coal pillars. Figure 13a also shows the source location and focal mechanism solution of 50 strong MIEs in the roadway area of the Gaojiapu Coal Mine from August to October 2018. In the 50 strong MIEs, shear-type sources accounted for 46 times, and tensile-type sources only accounted for 4 times. In addition, the horizontal projection of the five rock bursts is located in the coal pillar area of the roadways, vertically located in the coal seam or adjacent to the coal seam. It is suggested that the coal pillar is prone to shear instability along the pre-existing primary fracture when subjected to high-stress loading. As seen in Figure 13b, the azimuth of 50 strong MIEs was mostly concentrated between 120~150° and 310~330°, and dip angles were mostly less than 60°. Combined with the geological condition of the roadways, it was seen that coal pillars were easily subjected to continuous shear failure along the dominant direction of stress under the extrusion of roof and floor under the high static load stress (large buried depth, isolated coal pillar, etc.) in coal pillar area and dense layout of roadways. Moreover, as shown in Figure 13c, the average value of source parameters except for the apparent stress of shear-type sources are higher than that of tensile types. It can be concluded that the rock burst risk of shear-type MIEs in the high-stress roadway is significantly higher than that of tensile types.

It follows from the above statement that there is a significant difference in the focal mechanism and source parameters, where the geological structure, stress environment, and focal layer have a prominent influence. These response characteristics provide a basis for the accurate identification of strong MIEs and dynamic risk prediction in underground coal mines.

## 5. Conclusions

Mining-induced earthquakes are one of the most common dynamic phenomena in underground coal mines. It is of great significance to understand the response characteristics of focal mechanisms and source parameters of the MIEs to predict and control dynamic disasters. An inversion method of relative moment tensor was proposed to obtain the focal mechanism solution. An inversion matrix was constructed based on the source group, and the identification method of source rupture type was optimized. The response laws of focal mechanism and source parameters of mining-induced earthquakes in three frequent rock burst areas were analyzed. There was a significant difference between MIEs in different geological and engineering backgrounds. Three main conclusions were drawn as follows:In the typical fold structure area, the tensile-type sources are located in the roof and coal seam, while the shear types are primarily located near the coal seam. In addition, there is a close relationship between the occurrence of the rupture surface and the source elevation. The difference in source rupture strength and stress adjustment between tensile types and shear types is negligible, while the disturbance scale of tensile types is distinct.In the deep buried fault area, the rupture of sources in the roof and coal are dominant tensile and shear types, respectively. The tensile types have higher source energy but lower seismic moment than shear types. The apparent volume of shear types is about three times for tensile types. The apparent stress of the tensile types is higher than that of the shear types, representing that the stress concentration still exists in the roof after the MIEs, but the stress near the faults could be effectively released.In the high-stress roadway pillar area, the primary fracture of coal pillars easily produces continuous shear ruptures along the dominant stress direction under the extrusion of roofs and floors, which leads to a dominated shear-type rupture. The source parameters (except apparent stress) of shear types are higher than that of tensile types and show a higher dynamic risk.

## Figures and Tables

**Figure 1 ijerph-19-07352-f001:**
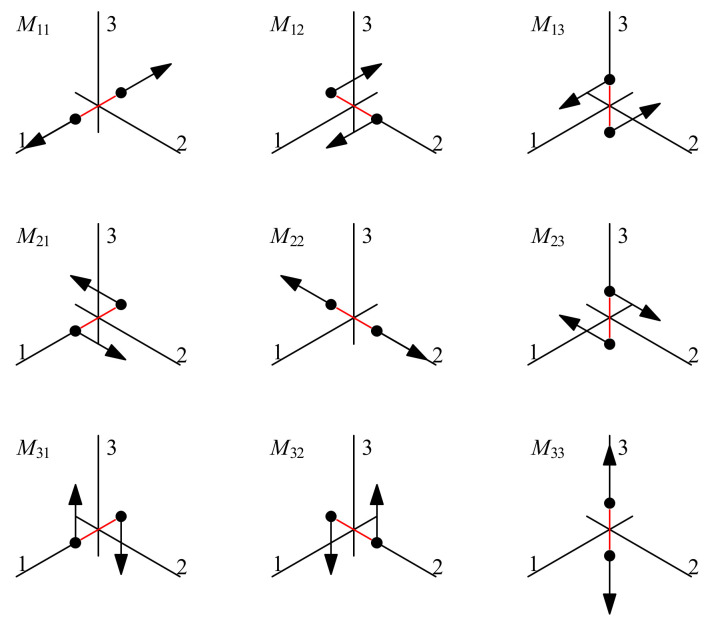
The nine different force couples that make up the moment tensor components. (After Shearer [43]).

**Figure 2 ijerph-19-07352-f002:**
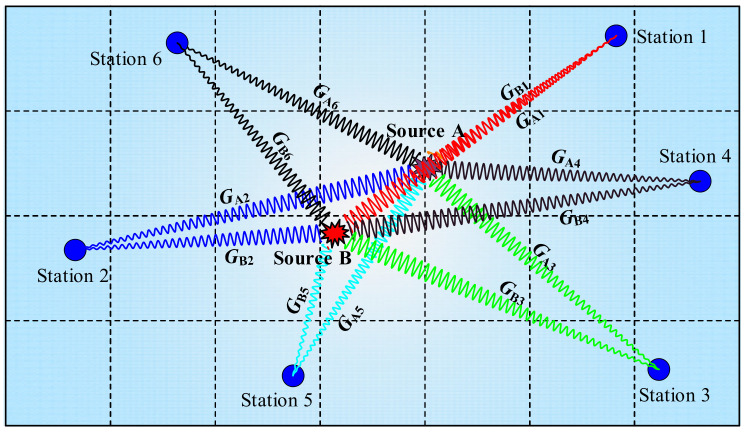
Schematic diagram of absolute MTI method. The ***G*_Ai_** or ***G*_Bi_** (*I* = 1 to 6, i.e., the sequence number of the stations) is the constructed Green’s function between arbitrary source and station. It is noticed that the stations in the coal mine move along with the excavation.

**Figure 3 ijerph-19-07352-f003:**
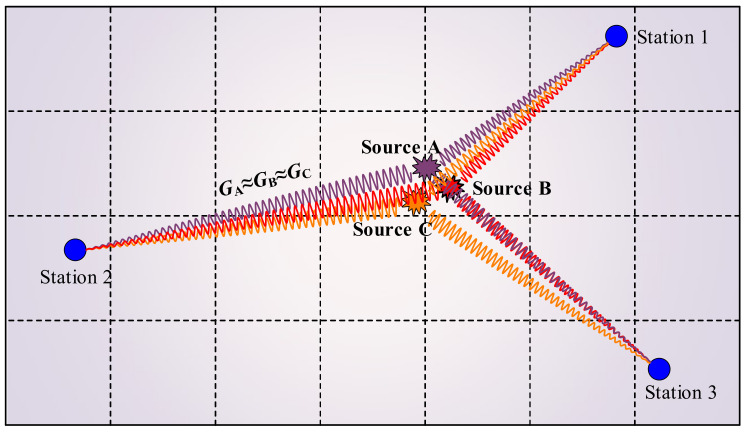
Schematic diagram of relative MTI method. The ***G*_A_**, ***G*_B,_** and ***G*_C_** represent similar Green’s functions among the source clusters. It is noticed that the stations in the coal mine move along with the excavation.

**Figure 4 ijerph-19-07352-f004:**
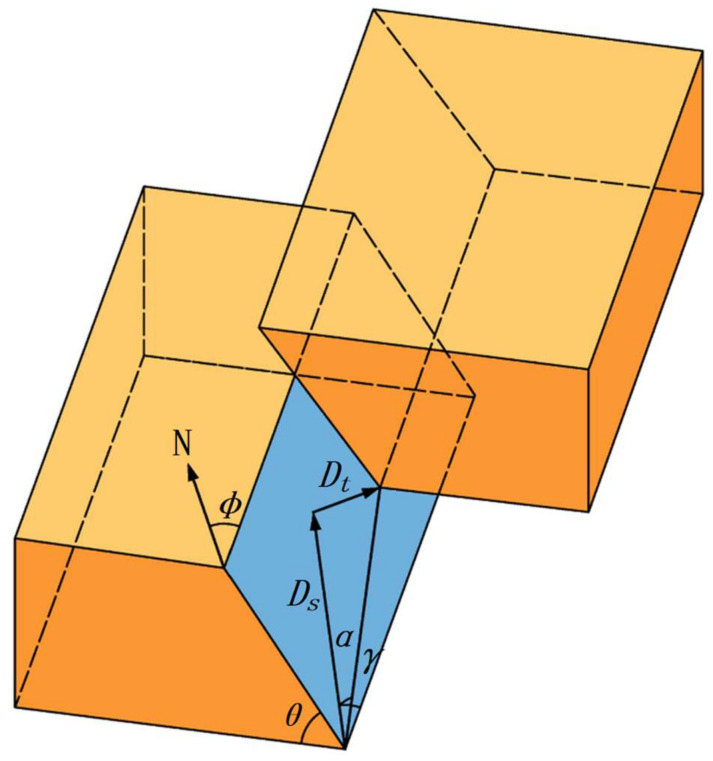
Schematic diagram of a general dislocation model for source rupture [59]. ***N*** represents the north direction; ***D_t_*** and ***D_s_*** represent the shear and vertical displacement vector, respectively; ϕ, θ, and γ represent the strike, dip, and slip angle, respectively; α represents the dislocation angle. In this fault model, the initial fault gap is not taken into account.

**Figure 6 ijerph-19-07352-f006:**
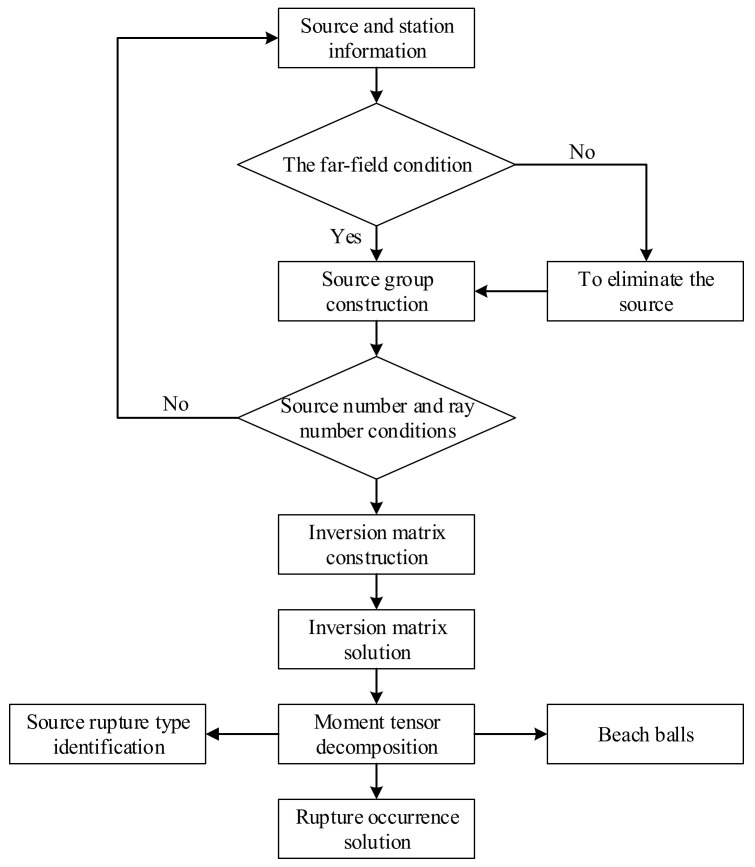
Flowchart for implementing the method to apply.

**Figure 7 ijerph-19-07352-f007:**
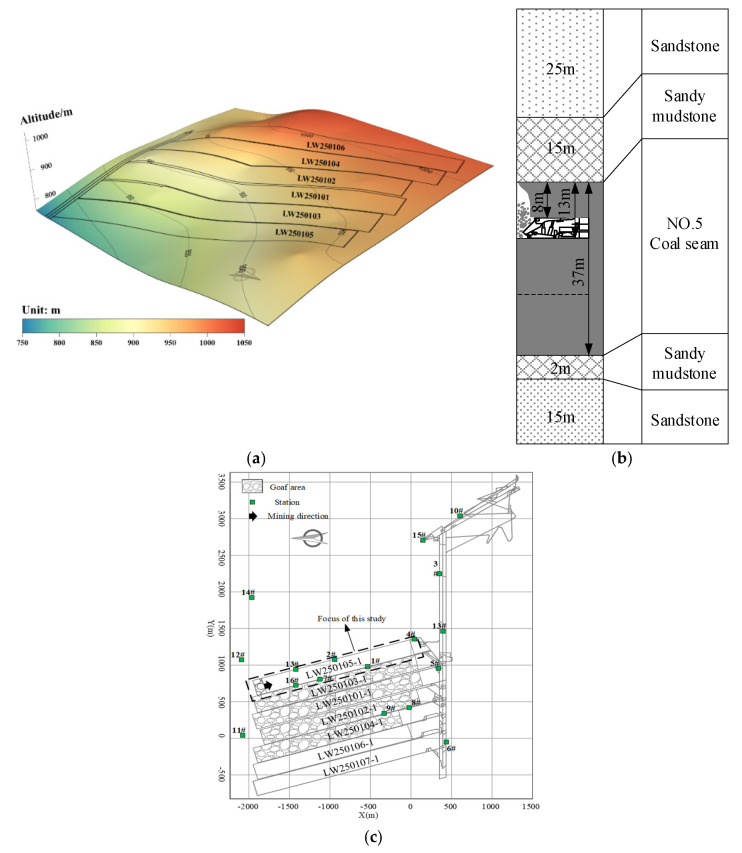
(**a**) LW250105−1 buried depth contour. (**b**) Stratigraphic column of LW250105−1. (**c**) LW250105−1 layout and station distribution.

**Figure 8 ijerph-19-07352-f008:**
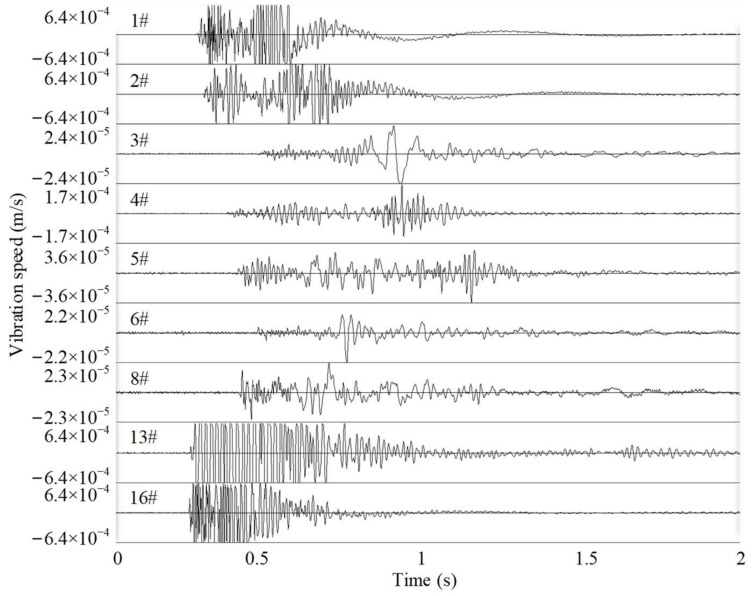
Typical seismic waves of a tensile type strong MIE in LW250105−1.

**Figure 9 ijerph-19-07352-f009:**
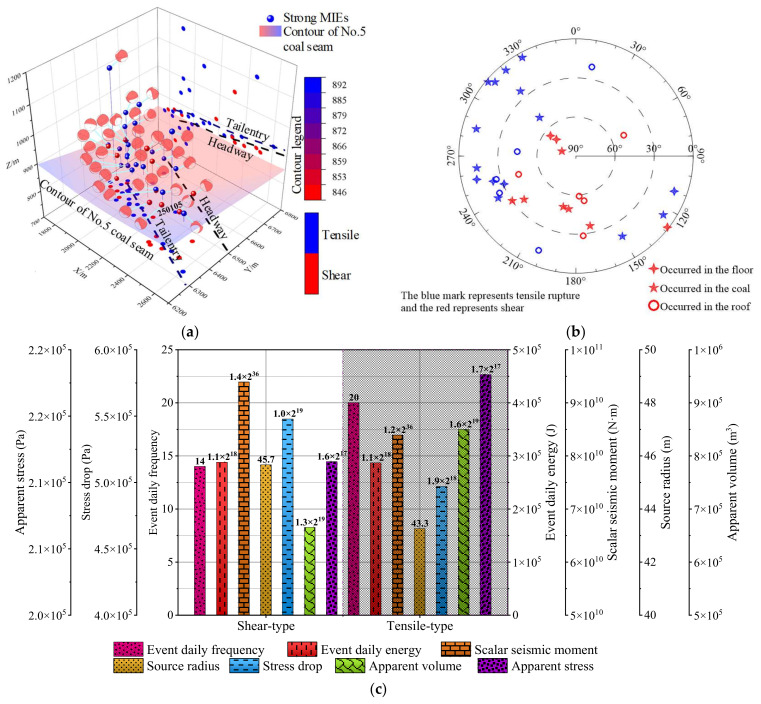
(**a**) Source location and focal mechanism solution of 34 strong MIEs during LW250105−1 retreat. (**b**) Source rupture occurrence of 34 strong MIEs. (**c**) Focal mechanism and response characteristics of focal parameters of 34 strong MIEs.

**Figure 10 ijerph-19-07352-f010:**
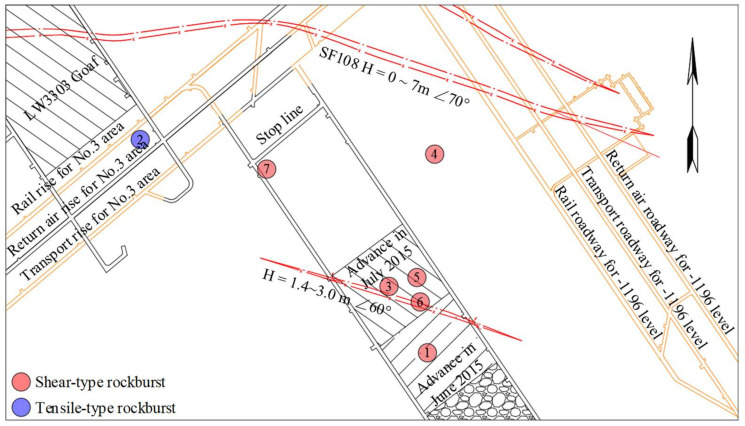
LW3302 layout and rock bursts distribution in Xingcun Coal Mine.

**Figure 11 ijerph-19-07352-f011:**
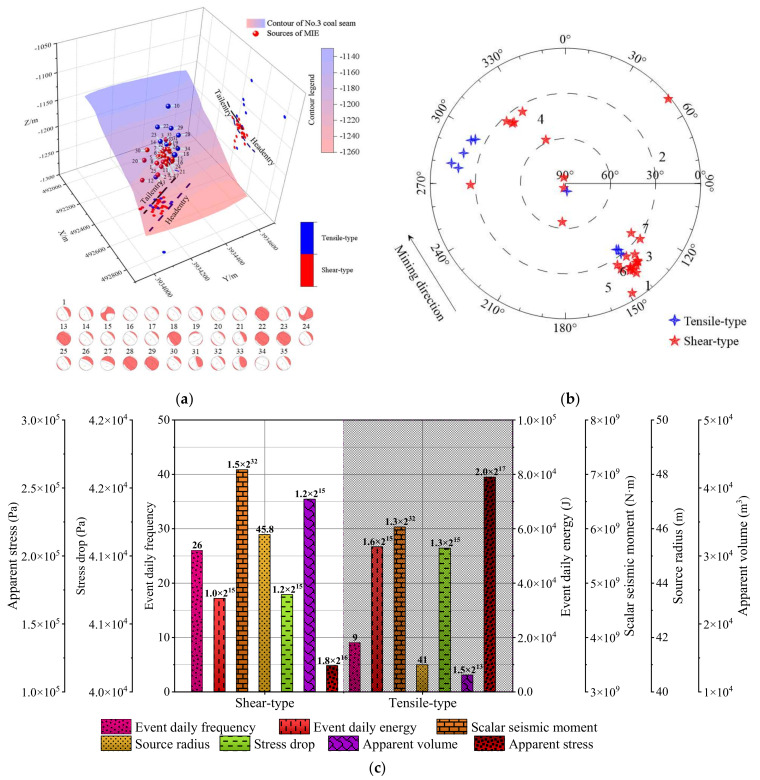
(**a**) Source location and focal mechanism solution of several strong MIEs during LW3302 retreat between June and July 2014. (**b**) Source rupture occurrence of strong MIEs. (**c**) Focal mechanism and response characteristics of focal parameters of strong MIEs.

**Figure 12 ijerph-19-07352-f012:**
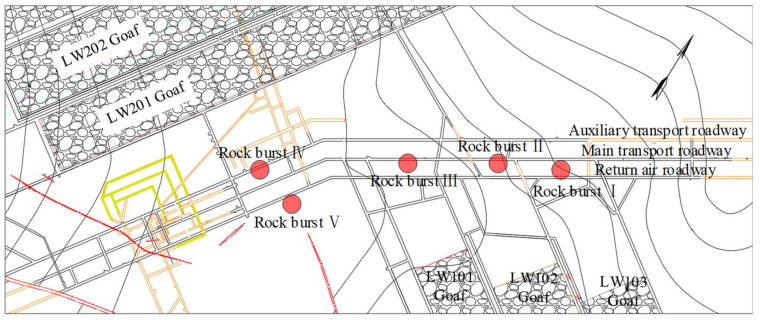
Roadway layout and rock bursts distribution in Gaojiapu Coal Mine.

**Figure 13 ijerph-19-07352-f013:**
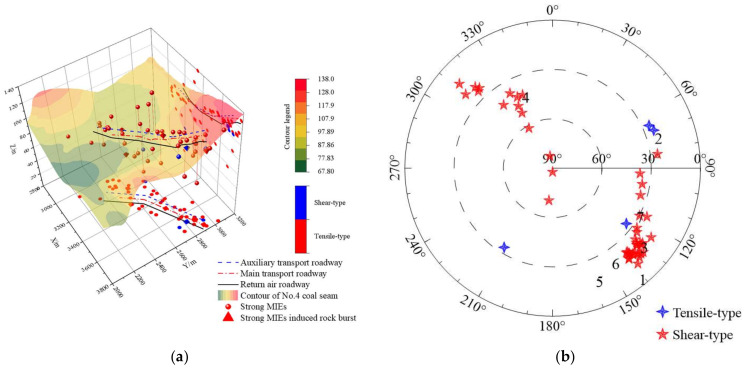
(**a**) Source location and focal mechanism solution of several strong MIEs in the roadway area of Gaojiapu Coal Mine. (**b**) Source rupture occurrence of strong MIEs. (**c**) Focal mechanism and response characteristics of focal parameters of strong MIEs.

## Data Availability

Not applicable.

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
