# Peer review of "Focal Mechanism and Source Parameters Analysis of Mining-Induced Earthquakes Based on Relative Moment Tensor Inversion"

_ijerph, 2022, doi:10.3390/ijerph19127352_

Round 1

Reviewer 1 Report

Overall work is good but one suggestion is about the length of the mathematical explanation. If possible, may reduce some detailed mathematical explanations to a reduced form. The author then can add the detailed work as appendix or supplement with the paper. So, those who want the detailed work can refer to it. 

Author Response

Response to Reviewer #1:

Comments and Suggestions for Authors:

Overall work is good but one suggestion is about the length of the mathematical explanation. If possible, may reduce some detailed mathematical explanations to a reduced form. The author then can add the detailed work as appendix or supplement with the paper. So, those who want the detailed work can refer to it.

Response: Thanks for the kind suggestion. We have transferred the detailed description of the relative method inversion matrix construction and solution method to the appendix.

Reviewer 2 Report

This is very good work.

The literature review is good and relevant.

The methodology part is laid out in a systematic manner and the relevant approaches are explained in detail. You could also have expanded a bit on the equations but this is ok, you have adequate.

The presentation of the results in appropriate and relevant. The illustrations used are impressive. I would also consider plotting contrasting curves on the same plot, but this is minor issue.

The conclusions are supported by the data.

Overall, great work, well done.

Author Response

Response to Reviewer #2:

Comments and Suggestions for Authors:

This is very good work.

The literature review is good and relevant.

The methodology part is laid out in a systematic manner and the relevant approaches are explained in detail. You could also have expanded a bit on the equations but this is ok, you have adequate.

The presentation of the results in appropriate and relevant. The illustrations used are impressive. I would also consider plotting contrasting curves on the same plot, but this is minor issue.

The conclusions are supported by the data.

Overall, great work, well done.

Response: Thanks for your recognition of our work. We will try our best to improve and refine our research work to expand the understanding of rock burst mechanisms.

Reviewer 3 Report

The issue presented in the article is very interesting because of the fact that the threat of induced seismicity is one of the most important issues occurring in many coal basins in the world. It is the subject of a number of studies.   However, it is difficult to agree with the authors' statement that the rupture  mechanism of MIE's sources and the relations between source and bump parameters are not systematically investigated. They are the subject of a number of studies.  

The most important part of the reviewed article is the presented results of research in three mines, which may contribute to better understanding of the focal mechanism and source parameters of mining seismic phenomena.

It should also be noted that the problem of prediction and control of the dynamic risk itself is very difficult, so the results presented here may help in predicting the potential hazard, which, of course, may help in making decisions regarding the use of appropriate methods to reduce the seismic risk.

Remarks:

Some ambiguity is raised by the description, concerning the Huating mine. The authors write about 34 rock bursts that occurred between March 2014 and May 2015. In Figure 8 (c) they present results for daily seismic energy and daily frequency. Similar uncertainties, relate to Figures 10 (c) and 12 (c). 

In the case of Figure 7, it might be more beneficial to include information on the types of seismic waves.

All symbols used in Figure 4 should be explained in the text.

Editing errors have been noted

Author Response

Response to Reviewer #3:

Comments and Suggestions for Authors:

The issue presented in the article is very interesting because of the fact that the threat of induced seismicity is one of the most important issues occurring in many coal basins in the world. It is the subject of a number of studies. However, it is difficult to agree with the authors' statement that the rupture mechanism of MIE's sources and the relations between source and bump parameters are not systematically investigated. They are the subject of a number of studies.

The most important part of the reviewed article is the presented results of research in three mines, which may contribute to better understanding of the focal mechanism and source parameters of mining seismic phenomena.

It should also be noted that the problem of prediction and control of the dynamic risk itself is very difficult, so the results presented here may help in predicting the potential hazard, which, of course, may help in making decisions regarding the use of appropriate methods to reduce the seismic risk.

Response: Thanks for your helpful suggestion. We focus on the response laws of focal mechanism and source parameters in frequent rock burst areas. We aim to find some common laws in different structural mining areas. In our future research, we plan to conduct a detailed investigation of every rock burst events combined with focal mechanism, source parameters, field measurement data, etc.

Remarks:

(1) Some ambiguity is raised by the description, concerning the Huating mine. The authors write about 34 rock bursts that occurred between March 2014 and May 2015. In Figure 8 (c) they present results for daily seismic energy and daily frequency. Similar uncertainties, relate to Figures 10 (c) and 12 (c).

Response: Thanks for the useful suggestion. We have unified all descriptions as strong mining induced earthquakes (MIEs).

(2) In the case of Figure 7, it might be more beneficial to include information on the types of seismic waves.

Response: Thanks for the kind suggestion. We have add the information of the types of seismic waves.

(3) All symbols used in Figure 4 should be explained in the text.

Response: Thanks for your useful suggestion. We have add the explanation of all symbols in the figure caption.

(4) Editing errors have been noted.

Response: Thanks for your kind suggestion. We have carefully checked the whole manuscript. Additionally, this revised manuscript has been modified by an English professional editor for language editing. The specific revisions were marked in the annotated version of the revised manuscript ("Marked Manuscript").

Reviewer 4 Report

Among the natural hazards occurring in the rock mass, the most dangerous are rock bursts. Despite the fact that scientific research is still carried out, the exact determination of the time of the discharge of seismic energy in the form of bumps has not been developed. The presented issu is important from the point of view of better identification of seismic phenomena occurring in underground hard coal mining. The valuable part of the article is the identification of the energy source and the division into tensile, shear and locations of dynamic phenomena in the floor, roof and coal. Below are some comments and suggestions:

1. In the introduction, a few sentences should be written regarding the numerical modeling with which it is possible to estimate the areas at risk of rock bursts.

2. In subsection 2.1, line 133, the sentence "... the minimum displacement is proposed more than 500 m in underground mining ...", it should be written the reason for this value of 500 m.

3. For Figure 4, it should be stated whether the calculation takes into account the width of the fualt gap.

4. In subsection 4.1 it is necessary to write what is the width of the carbon pillar left between adjacent longwall panel.

5. In the summary, one conclusion should be written regarding the numerical values presented in Figure 12, in particular with regard to the statement where the most bumps occur in front of or behind the mine front.

Author Response

Response to Reviewer #4:

Comments and Suggestions for Authors:

Among the natural hazards occurring in the rock mass, the most dangerous are rock bursts. Despite the fact that scientific research is still carried out, the exact determination of the time of the discharge of seismic energy in the form of bumps has not been developed. The presented issue is important from the point of view of better identification of seismic phenomena occurring in underground hard coal mining. The valuable part of the article is the identification of the energy source and the division into tensile, shear and locations of dynamic phenomena in the floor, roof and coal. Below are some comments and suggestions:

(1) In the introduction, a few sentences should be written regarding the numerical modeling with which it is possible to estimate the areas at risk of rock bursts.

Response: Thanks for the kind suggestion. The sentences regarding the numerical modeling with which it is possible to estimate the areas at risk of rock bursts has been added to the introduction (Line 57-60 in the marked manuscript). The statement is also shown as follows:

“Therein, the numerical modeling also sheds light on the application on assessing rock burst risk [1-6]. For example, the stress, displacement, and energy evolution characteristics along the rock burst process can be investigated in detail.”

(2) In subsection 2.1, line 133, the sentence " the minimum displacement is proposed more than 500 m in underground mining ", it should be written the reason for this value of 500 m.

Response: Thanks for the useful suggestion. The reason for this value of 500 m has been added in the context (Line 133-136 in the marked manuscript). The description is also shown as follows:

“Where u is far-field displacement sequence matrix obtained from different stations, which means only if the displacement sequence obtain from obtained from stations far enough from the source the Eq. (4) can be constructed, and the minimum displacement condition is proposed as more than 500 m in underground mining.”

(3) For Figure 4, it should be stated whether the calculation takes into account the width of the fault gap.

Response: Thanks for the kind suggestion. The statement has been add in the caption of the Fig. 4 (Line 224 in the marked manuscript). The statement is also shown as follows:

“In this fault model, the initial fault gap is not taken into account.”

(4) In subsection 4.1 it is necessary to write what is the width of the carbon pillar left between adjacent longwall panel.

Response: Thanks for the kind suggestion. The description has been added in the marked manuscript (Line 321 – 322), which is shown as follows:

“There are 6m rib block coal pillars between longwall panels to isolate the goaf.”

(5) In the summary, one conclusion should be written regarding the numerical values presented in Figure 12, in particular with regard to the statement where the most bumps occur in front of or behind the mine front.

Response: Thanks for the kind suggestion. We have added the statement about the numerical values presented in Fig. 12 and draw some conclusions as seen in the marked manuscript (Line 448 – 467), which is also shown as follows:

As seen in Fig. 12 and Fig. 13 (a), the coal pillars of the roadway bear a high abutment pressure due to the goaf area around the roadway. In addition, the repeated disturbance of the longwall mining leads to the development of primary fractures in the coal pillars, leading to a decrease in the stability of the coal pillars. Fig. 13 (a) also shows the source location and focal mechanism solution of 50 strong MIEs in the roadway area of Gaojiapu Coal Mine from August to October 2018. In the 50 strong MIEs, shear type sources accounted for 46 times, and tensile type sources only accounted for four times. In addition, the horizontal projection of the five rock bursts is located in the coal pillar area of the roadways, vertically located in the coal seam or adjacent to the coal seam. It’s suggested that the coal pillar is prone to shear instability along the pre-existing primary fracture when subjected to high-stress loading. As seen in Fig. 13 (b), the azimuth of fifty strong MIEs mostly concentrate between 120°~150° and 310°~330°, and dip angles are mostly less than 60°. Combined with the geological condition of roadways, it was seen that coal pillar is easy to be subjected to continuous shear failure along the dominant direction of stress under the extrusion of roof and floor under the high static load stress (large buried depth, isolated coal pillar, etc.) in coal pillar area and dense layout of roadways. Moreover, as shown in Fig. 13 (c), the average value of source parameters except for the apparent stress of shear type sources are higher than that of tensile types. It’s concluded that the rock burst risk of shear type MIEs in high-stress roadway is significantly higher than that of tensile types.”

Reviewer 5 Report

The study of focal mechanism of mining events is a meaningful and frontier research, the authors proposed a relative moment tensor inversion method and applied it in a mine. It is a very good research with high quality figures, however, there are some unclear places needing clarification before it can be accepted.

Minor revisions

1. Introduction: more references should be included to introduce the moment tensor inversion method.

2. Section 2: The method introduction is hard to understand. As an expert in this field, I don’t know how did you do the “relative” procedure for the relative moment tensor inversion.

3. How to transform equation (10) to (11), what is the meaning of f and u in equation (11)? In other words, how to confirm the A in equation (11)?

4. Are the meaning of G in equations (4) and (21) are different?

5. A detailed example had better be included to show how to implement the method to application in section 4.

6. Fig. 7: The signal has poor P, SH and SV amplitudes, which will affect the inversion result of focal mechanism, what is the reliability of the inversion results? 

Author Response

Response to Reviewer #5:

Comments and Suggestions for Authors:

The study of focal mechanism of mining events is a meaningful and frontier research, the authors proposed a relative moment tensor inversion method and applied it in a mine. It is a very good research with high quality figures, however, there are some unclear places needing clarification before it can be accepted.

Minor revisions:

  1. Introduction: more references should be included to introduce the moment tensor inversion method.

Response: Thanks for the kind suggestion. We have added the description of moment tensor inversion method in the introduction (Line 70-79 in the marked manuscript). The description is also shown as follows:

“Recently years, several MTI methods have been developed through fitting different kinds of observations and source models, including seismic waveforms [7], first motion polarities [8], P- and S-wave amplitude ratios [9], the combination of spectral amplitudes or amplitude ratios with P-wave polarities [10] and full-waveform method [11], etc. Therein, the accurate construction of Green’s functions is necessary. However, as the MTI of these methods is influenced by station layouts, focal mechanism, medium properties, etc., when performing on MIEs in coal mines [12], the accurate construction of Green’s functions is complicated. Therefore, a modified MTI method applying to MIEs in underground coal mines should be explored to understand the source mechanism of MIEs.”

  1. Section 2: The method introduction is hard to understand. As an expert in this field, I don’t know how did you do the “relative” procedure for the relative moment tensor inversion.

Response: The biggest difference between the relative method and the absolute is reflected in the treatment of Green's function. In the absolute methods, the Green’s functions are evaluated theoretically using an appropriate earth model and algorithm, or inferred empirically from observations of a known source. One of the difficulties in the absolute inversion (apart from noise affecting the input data) is the accurate estimation of the Green’s functions for geologically complex media.

In contrast to the absolute methods, the relative inversion methods do not require the calculation of theoretical Green’s functions for each event. Relative methods are based on the concept of a common ray-path between a cluster of seismic sources and any receiver, and assume that all the events in the cluster experience the same wave propagation effects to each receiver. Generally, the radiation pattern of a reference event is used to estimate the Green’s functions for events from the same source region. In our paper, we use the linear wave propagation terms () and radiation patterns () to express the response characteristics of the medium during wave propagation (seen Line 485-490 in the marked manuscript). In addition, in order to eliminate the common ray-path terms (), we constructed Eq. (14), and the inversion matrix can be further formed. Especially in our method, the every source can be used to construct inversion matrix (seen the Eq. (20) in the marked manuscript).

  1. How to transform equation (10) to (11), what is the meaning of f and u in equation (11)? In other words, how to confirm the A in equation (11)?

Response: We added the detailed mathematical explanation of relative MTI on the Appendix A proposed by Reviewer #1. In addition, we rewrote the equation (10) to (11) of the origin manuscript to make readers understand easier. The revised description can be seen in the Appendix A (Line 504-516 in the marked manuscript).

  1. Are the meaning of G in equations (4) and (21) are different?

Response: The meaning of G in equations (4) and (21) is consistent. They represent the response characteristics of the medium during wave propagation.

  1. A detailed example had better be included to show how to implement the method to application in section 4.

Response: Thanks for the kind suggestion. We have added the guidelines on the implementation (seen in section 3.3 of the marked manuscript).

  1. Fig. 7: The signal has poor P, SH and SV amplitudes, which will affect the inversion result of focal mechanism, what is the reliability of the inversion results?

Response: Thanks for the useful suggestion. The Fig. 7 shows the typical seismic waves of the one of strong MIEs. The signal has poor P, SH and SV amplitudes in some panels, e.g. panel 1#, 13# and 16# in Fig. 8. However, in the moment tensor inversion, these panels are not used and only the panels 2#, 3#, 4#, 5#, 6# and 8# were used. In addition, in order not to confuse the readers, a description of the waveform selection has been added to the detailed implement example in section 4.1 (Line 349-352 in the marked manuscript).

References

  1. Cao, A.; Liu, Y.; Jiang, S.; Hao, Q.; Peng, Y.; Bai, X.; Yang, X. Numerical Investigation on Influence of Two Combined Faults and Its Structure Features on Rock Burst Mechanism. MINERALS 2021, 11, doi:10.3390/min11121438.
  2. Cao, W.; Shi, J.; Si, G.; Durucan, S.; Korre, A. Numerical modelling of microseismicity associated with longwall coal mining. International Journal of Coal Geology 2018, 193, 30-45, doi:https://doi.org/10.1016/j.coal.2018.04.010.
  3. Keneti, A.; Sainsbury, B.; Dargaville, R. Consideration of Strain-Bursting Phenomena Associated with Large-Scale Discontinuities: A Numerical Study. Pure and Applied Geophysics 2021, 178, 3581-3600, doi:10.1007/s00024-021-02851-7.
  4. Manouchehrian, A.; Cai, M. Numerical modeling of rockburst near fault zones in deep tunnels. Tunnelling and Underground Space Technology 2018, 80, 164-180, doi:10.1016/j.tust.2018.06.015.
  5. Sainoki, A.; Schwartzkopff, A.K.; Jiang, L.; Mitri, H.S. Numerical Modeling of Complex Stress State in a Fault Damage Zone and Its Implication on Near-Fault Seismic Activity. Journal of Geophysical Research: Solid Earth 2021, 126, e2021JB021784, doi:https://doi.org/10.1029/2021JB021784.
  6. Vardar, O.; Zhang, C.; Canbulat, I.; Hebblewhite, B. Numerical modelling of strength and energy release characteristics of pillar-scale coal mass. Journal of Rock Mechanics and Geotechnical Engineering 2019, 11, 935-943, doi:https://doi.org/10.1016/j.jrmge.2019.04.003.
  7. Vavrycuk, V.; Kühn, D. Moment tensor inversion of waveforms: A two-step time-frequency approach. Geophysical Journal International 2012, 190, 1761, doi:10.1111/j.1365-246X.2012.05592.x.
  8. Nakamura, M. Determination of focal mechanism solution using initial motion polarity of P and S waves. Physics of the Earth and Planetary Interiors 2002, 130, 17-29, doi:10.1016/S0031-9201(01)00306-5.
  9. Jechumtálová, Z.; Šílený, J. Amplitude ratios for complete moment tensor retrieval. Geophysical Research Letters 2005, 32, doi:10.1029/2005GL023967.
  10. Li, J.; Zhang, H.; Kuleli, H.; Toksoz, M. Focal Mechanism Determination Using High Frequency Waveform Matching and Its Application to Small Magnitude Induced Earthquakes. Geophysical Journal International 2011, 184, 1261-1274, doi:10.1111/j.1365-246X.2010.04903.x.
  11. Ma, J.; Dineva, S.; Cesca, S.; Heimann, S. Moment tensor inversion with three-dimensional sensor configuration of mining induced seismicity (Kiruna mine, Sweden). Geophysical Journal International 2018, 213, doi:10.1093/gji/ggy115.
  12. Aki, K.; Richards, P. Quantitative Seismology Theory. W.H. Freeman: San Francisco, 2002; Volume vol. I.